# Design of Refractory Alloys for Desired Thermal Conductivity via AI-Assisted In-Silico Microstructure Realization

**DOI:** 10.3390/ma16031088

**Published:** 2023-01-27

**Authors:** Seyed Mohammad Ali Seyed Mahmoud, Ghader Faraji, Mostafa Baghani, Mohammad Saber Hashemi, Azadeh Sheidaei, Majid Baniassadi

**Affiliations:** 1School of Mechanical Engineering, College of Engineering, University of Tehran, Tehran 15614, Iran; 2Aerospace Engineering Department, Iowa State University, Ames, IA 50011, USA

**Keywords:** microstructure identifications, machine learning, thermal characterization, Fast Fourier Transform (FFT) homogenization

## Abstract

A computational methodology based on supervised machine learning (ML) is described for characterizing and designing anisotropic refractory composite alloys with desired thermal conductivities (TCs). The structural design variables are parameters of our fast computational microstructure generator, which were linked to the physical properties. Based on the Sobol sequence, a sufficiently large dataset of artificial microstructures with a fixed volume fraction (VF) was created. The TCs were calculated using our previously developed fast Fourier transform (FFT) homogenization approach. The resulting dataset was used to train our optimal autoencoder, establishing the intricate links between the material’s structure and properties. Specifically, the trained ML model’s inverse design of tungsten-30% (VF) copper with desired TCs was investigated. According to our case studies, our computational model accurately predicts TCs based on two perpendicular cut-section images of the experimental microstructures. The approach can be expanded to the robust inverse design of other material systems based on the target TCs.

## 1. Introduction

Computational material design, a fast-growing research field, is promising for discovering sophisticated multifunctional materials. The success of this line of research is contingent on the proper representation of the material microstructure. Microstructure characterization and reconstruction (MCR) techniques are divided into five categories: (1) correlation function-based methods [1,2,3,4]; (2) physical descriptor-based methods [5,6,7,8]; (3) spectral density function-based characterization and reconstruction by level-cutting a random field [9] or disk-packing [10]; (4) ML-based methods such as convolutional deep neural networks [11], instance-based learning [12], and encoding/decoding methods [13]; and (5) texture synthesis-based methods [14,15,16]. Because they lack specific or physical design characteristics, categories 1, 4, and 5 cannot be employed for material design. Others may involve dimensional reduction due to high-dimensional representations [17], which should be carefully investigated to avoid severe information loss and reduce structural variability. Overall, the physical descriptor-based method is the most convenient and capable category for material design [18]. Dealing with massive data, ML approaches were employed to learn the intricate relationship between microstructure characteristics and their homogenized responses. For instance, Hashemi et al. [19,20,21] recently developed unique ML-based computational frameworks for homogenizing and designing heterogeneous soft materials. In addition, Bessa et al. [22] have suggested a framework for data-driven analysis and material systems design in the face of uncertainty. Such material science studies are intrinsically complex for various reasons, including the challenges of resolving heterogeneities in the material, non-linearity of the material’s reaction and boundary conditions, and excessive dimensionality of the design space, as documented in such frameworks. Reduced-order models (ROMs) could be used to speed up the data creation process. Such developments have been the subject of several scientific studies. Liu et al. [23] devised a self-consistent clustering methodology to anticipate irreversible processes reliably.

In this study, we focus on the computational design of tungsten–copper composites. Tungsten is one of the densest metals with the highest melting point in the periodic table. In tungsten-based composites, the weight percentage of tungsten ranges from 60 to 98 percent. As a secondary phase, they contain a mixture of low-melting-temperature metal components, such as copper, iron, cobalt, and nickel. The density of these metal composites is close to one (i.e., very low void VF), with excellent toughness, flexibility, and dimensional stability [24]. The tungsten–copper composite is the most prevalent tungsten-based composite. Tungsten has a high melting temperature, excellent wear resistance, corrosion resistance, and a low thermal expansion coefficient. In contrast, copper has high flexibility, excellent thermal and electrical conductivity, and excellent forming properties [25]. Therefore, tungsten–copper composites have unique features, such as high strength and hardness, high corrosion and wear resistance, a low thermal expansion coefficient, high flexibility, outstanding thermal and electrical conductivity, and good arc wear resistance [24,25]. They also have many applications in the electronics, military, aerospace, and thermal industries, including high-voltage power contactors, rocket engine nozzles, and other related applications [22,24,26]. In these composites, tungsten is the hard phase, and dislocations cannot pass through it, thereby forcing them to bypass it and enhancing the material’s strength [27]. Furthermore, due to tungssten’s high modulus of elasticity, the higher the amount of tungsten is in the composite, the higher the compressive strength it has. On the other hand, the smaller the copper content is in the composite, the more likely cracks between tungsten particles will emerge during deformation. Due to the penetration of molten copper into the tungsten skeleton, the sintered microstructure becomes more homogeneous as the amount of copper increases, and the underlying performance against crack formation improves. The toughness and flexibility of these composites are heavily influenced by the bond strength between the phases. Flexibility improves as the copper phase’s binding strength increases [28].

We tried to maximize the thermal property of tungsten composites with 20wt% or 30% VF copper [29,30] through the microstructure design of this material system. In recent decades, there has been a lot of research in determining the functional qualities of composite materials based on their unique ingredients. High-fidelity finite element (FE) simulations of composite materials’ behavior provide accurate predictions, but their utility in the design process is limited by their demanding computational time. Using optimum ML methods, we built a computational framework to obtain accurate and affordable predictions of TCs of tungsten–copper composites and understand the influence of microstructural geometry on them. Figure 1 depicts a high-level overview of this material discovery framework. To train ML models on a labeled dataset, the first phase is data generation if there is not enough available data. Phase 2 entails utilizing appropriate ML methods to discover the intricate relationships between the microstructure and properties. Recently, material science researchers are increasingly using machine learning methods to predict properties of materials [31,32]. Most of these algorithms have not been used for predicting heterogenous materials’ properties. Also, the maximum accuracy of them was 90% [33], and in comparison with our studies, they have enormous computational costs. To predict the mechanical properties of composites, backpropagation algorithms are widely used due to their high convergent precision. The database constructed for thermal conductivity properties was developed using a method based on the fractal system [34]. Considering that two-point correlation functions have a direct relationship with thermal properties [35] and the fractal algorithm used has the ability to create microstructures with different two-point correlation functions, the use of the fractal algorithm provides us with a comprehensive set of microstructures. For homogenization, the FFT method was utilized to calculate the properties for the eigen microstructure directly.The first goal of this research is to determine the material system’s effective properties based on the microstructural parameters used in our microstructure generator code. The second goal is the inverse design or determining microstructure parameters given target properties as inputs. The third goal is to develop software to estimate the TCs based on two perpendicular cut-section images of the experimentally obtained microstructures. The microstructure could be produced using our developed procedures for the studied material system, given its parameters or 3D visualization. As a result, Phase 3 includes deducing direct microstructure–property correlations, as well as developing and visualizing candidate microstructures using the inverse design framework. The computational times provided in this study are based on a machine with an AMD RYZEN 5 2600X 3.6 GHz 19 MB BOX CPU and 32 GB DDR4 RAM. Due to the percolation phenomena, predicting TC based on the cut-section images is challenging. Thus, we have developed a method based on image processing, two-point correlation functions (TPCFs), and our homogenized dataset of artificially created microstructures to estimate the TCs of a given experimental composite.

## 2. Methods

### 2.1. Data Generation

We cannot just rely on the material’s experimental results to provide a large enough dataset for sophisticated ML algorithms. Even the most well-designed experimental approaches cannot cover all feature vectors or material system features. In this case, it is necessary for ML training to be as diverse and representative as its computational equivalents [36]. To have a representative dataset of the random variables, the physical microstructural descriptors in this work, which affect the performance of the trained ML model, a robust and efficient design scheme of the virtual experiments, i.e., computational simulations, was required. This step produces an open-source dataset that has the potential to expand the scope of material discovery research, particularly when it can be used for similar material systems of composites with diverse constituents.

#### 2.1.1. Realization of the Set of Microstructures

According to our previously developed methods, a set of microstructures was generated. The detail of the approach has been explained in our previously published work [20]. We considered seven input factors for realizing microstructures, including the number of phases, VFs of phases, and other design parameters, which are the input parameters of the artificial microstructure generation algorithm. In this approach, in the first step, a predetermined number of seeds were dispersed for each phase, and in the second step, the seeds grew according to the input parameters (growth rates). The cellular automata approach was exploited to model seed growth. Figure 2 shows the algorithm for generating microstructures. In Table 1, the range and values of the input parameters are summarized. The seed growth rate in the X and Y directions differed from Z’s to realize anisotropic microstructures because the manufactured composite is orthotropic. A sample of our artificially generated microstructures is shown in Figure 3.

A design of experiment (DoE) method was utilized to explore the design or input variables’ domain for efficiently training the ML model after identifying and limiting the microstructure factors impacting the characteristics. Because the conditional probabilities of the microstructural inputs and property outputs were unknown prior to the design, space-filling designs that equally cover all regions of the design space were chosen. A Sobol sequence [37], a deterministic low discrepancy sequence, and Latin hypercube samplings [38] have shown a better balance between more regular distribution or randomness and are closer to a regular grid or greater coverage of the input variables space [39]. As a result, we chose the Sobol sequence, which generates experiment points quickly.

#### 2.1.2. Calculating TC of the Generated Microstructures

An effective or homogenized TC tensor of the composite completes our dataset for material design and characterization. A homogenization technique was used to measure the effective TC. Because the effective TC [33,34] of a microstructure can be determined using the linear PDE equation of heat conduction and constant material coefficients, homogenization may be performed using a set of constant periodic boundary conditions. Consequently, the microstructural morphology is the only remaining feature that influences the composite property. Therefore, we used our previously developed FFT approach, an efficient algorithm for TC calculation, on our generated microstructures with diverse morphologies.

Copper and tungsten were given TC values of 0.25 W/mK and 174 W/mK, respectively. Traditional numerical approaches for determining the effective characteristics of random heterogeneous materials [40,41], such as FE methods, are limited by their reliance on very tiny and high-quality mesh adhering to detailed phase geometries. With voxelized representative volume elements (RVE), the FFT technique is demonstrated to be efficient because no conformal meshing is required [42]. It also outperforms other numerical methods in terms of scalability, with a complexity of O(NlogN) vs. $O\left(N^{3}\right)$, where N is the number of discretization grid points or voxels per dimension. We have validated our homogenization method with experimental findings in a separate study [43], and the reader is directed to that work for further information.

The level of details in the microstructures’ morphologies is limited by our realization resolution. The resolution, i.e., N, cannot be raised arbitrarily because the cost of FFT homogenization scales super-linearly with it. We initially set N to be equal to 100 in all directions, resulting in an average FFT computation time of 30 min per microstructure. However, it is necessary to investigate whether the generated microstructures are RVEs. Therefore, we studied the size effect on the homogenized property by realizing several higher-resolution microstructures and homogenizing them via the same FFT method. The RVEs with one million voxels capture the property accurately. As a result, we picked the lowest resolution required to reduce the computational cost of data generation in the FFT homogenization stage. Table 2 shows a comparison of TCs between the 200-voxel and 100-voxel RVEs given the same seed growth probabilities in each case.

### 2.2. ML Model Training

The goal of Phase 2 is to develop an efficient and optimal ML model to replace the time-consuming homogenization process. As a regression technique for modeling complex functions, neural networks are adaptable and robust [44]. Each neuron can be a nonlinear function, and the complexity of the system can be increased arbitrarily by varying network designs, the number of neurons, the number of layers, and the links between neurons. As a result, we evaluated various designs and used the 5-fold cross-validation technique to train them on the dataset. Different activation functions, such as ReLU and Sigmoid, were investigated once the data had been normalized, with ReLU proving to be the most successful. In the bottleneck of the autoencoder, as described below, we considered vectors of physical descriptors and other realization parameters. The number of input and output parameters must be supplied before the layers can be determined. The frequency of seed addition and seed growth rate in two main directions constitute the three-dimensional microstructural vectors. The property vectors have eight parameters representing the thermal and mechanical properties found by our FFT code. Specifically, three of them are the three diagonal components of the homogenized TC tensors, and the rest are five distinct components of the homogenized and mechanical stiffness tensors for orthotropic materials. The entire homogenized dataset was randomly divided into five equal-sized sets. The neural networks were trained five times using a portion of data that had not previously been considered as a test set and the rest as a training set each time. The average training accuracy and standard deviation were calculated after five training cycles so that the performance of different network architectures could be compared. For the final training on the entire dataset, the best-performing network with the lowest average training error (mean square error) was chosen.

### 2.3. Inferring a Complex Microstructure–Property Relationship

Based on Section 2.2, a fast and reliable ML model for predicting material properties can be constructed to operate as a surrogate for relatively expensive direct numerical solvers and to create a direct relationship between the microstructure of the studied material system and its effective homogenized properties. The more difficult topic is inverse design, which has proven difficult due to inefficient and expensive ways of determining the optimum material structure given the desired or target properties, particularly when dealing with the sophisticated characterization of microstructure images with too many features. For inverse design, a modified autoencoder was applied in this study. The best architecture selected in the previous step was considered to establish the structure–property links, where the property vectors are provided at both ends of the autoencoder. The new strategy employed in this study is to transform the latent space into a meaningful space by considering the microstructural and physical descriptors and their contributions to the total loss function. Figure 4 depicts the architecture of the inverse material design optimization approach. In this network, the trained encoder can be used as a fast inverse material designer, and the trained decoder can be used as a fast surrogate of numerical homogenizations.

## 3. Results and Discussion

### 3.1. Generated Dataset

We created 1000 data entries or microstructures after defining the primary factors for microstructure creation and selecting the design of the experiment method. Space-filling designs should almost uniformly cover the design space while maintaining non-collapsing limitations. The criteria for our Sobol DoE were met based on Figure 5 of the created realizations, albeit some sets of parameters were not used in the final simulations due to physical inconsistencies. This DoE scheme has the advantage of sequentially covering space while also generating sequences, allowing the dataset to be incrementally enhanced, i.e., the design space can be further explored by continuing earlier number sequences.

Figure 6 shows a 3D representation of a microstructure and a sample of its 2D slices (a black-and-white image).

### 3.2. Optimized Surrogate Model of Direct Structure–Property Relationship

Several fully connected neural network architectures were trained, evaluated, and compared to identify a network with high expected prediction accuracy, as indicated in Section 2. Table 3 summarizes their performances.

Table 3 and Figure 7a demonstrate the best network with the lowest mean squared error (MSE) using the cross-validation technique. The accuracy of large conductivity composites is reduced due to fewer DoE points covering parts of design space with greater growth rates, as shown in its regression plot across the entire dataset, Figure 7b. Furthermore, as shown in Figure 7b, most errors with respect to the homogenized realizations are minor. The conventional method of realization plus homogenization takes an average of 1–4 h for each microstructure in our generated dataset, while the speed of the surrogate model in terms of a trained neural network is on the order of 0.1 s.

### 3.3. Inverse Design via Modified Autoencoder

The trained encoder acts as the inverse design calculator (with no optimization effort for new inverse problems). The MSE performance on the test dataset was 3%, which shows its high accuracy. After several experiments, it was determined that the inverse design process is efficient and accurate. The inaccuracy could be due to insufficient data for ML training. The whole inverse design optimization took an average of 5 min. A summary of the computational times is shown in Table 4 to stress the effectiveness of our computational framework. Each design point in the inverse design optimization loop would have taken about 6 h to find the optimum material structure if there was no fast ML model. However, it only takes seconds with our trained model.

### 3.4. Connection of Computational Results to Real Process and Experiment

#### 3.4.1. Characterizing Real Manufactured Microstructures Using Its Cut-Section Images and Our Generated Computational Dataset

The algorithm developed for this part of our study is capable of characterizing real or experimentally manufactured tungsten–copper composites. In this method, two perpendicular optical images of the real microstructure are fed into our algorithm based on TPCFs and image processing. We first calculate two TPCFs of the given specimen (given its two perpendicular images) and then compare them with those of our computationally generated dataset containing 2 × 1000 photos. As shown in the top part of Figure 8, the specimen has been photographed in two perpendicular directions due to the anisotropic nature of the tungsten–copper microstructures. The maximum allowable error in TPCFs is 15% in our code, as shown in our algorithm flowchart in Figure 9. If a microstructure passes the TPCF similarity check, it will be further processed to determine its image features. If its image features resemble those of the real microstructure the most among all TPCF-passed microstructures, it will be identified as the closest microstructure in our generated dataset, and its properties can be retrieved from our dataset without any more computational cost. The identified microstructure in our dataset for the given experimental specimen of Figure 8 is visualized at the bottom of Figure 8, and it has a 4.87% error according to our image processing technique with its high-level overview shown in Figure 10.

More distinct specimens were manufactured to test our proposed method. The results are presented in Figure 11. The first column shows the cut-section images of the manufactured specimens, the second one visualizes the best-match computational microstructures found in our dataset via the above algorithm, and the last one presents the precalculated TC values (diagonal components of TC tensor) through our FFT homogenization performed in the data generation phase.

#### 3.4.2. VF and Defect Effects

This section illustrates why the considered VF for the copper, i.e., 30%, is desired for the studied material system. Different microstructures with different VF ratios of tungsten–copper were manufactured to investigate the amount of copper percolation in different volumetric fraction values, as shown in Figure 12. The vertical axis shows the length of the largest cluster divided by the microstructure size. Based on our experimental results, a copper VF larger than 30% is suitable for high clustering and better thermal properties due to the percolation effect.

The copper VF and the defects or voids, i.e., air or other inclusions, in the microstructure are the two most important factors affecting the thermal property in this composite. The processing method of composite manufacturing is to blame for defects. As illustrated in Figure 13, voids have a substantial impact on the thermal properties; the lower the percentage of voids in the structure is, the higher its thermal properties are. In this computational analysis, the copper VF was fixed at 30%, while the tungsten VF decreased as the void VF increased, and all microstructures were generated with equal growth rates in all directions to achieve isotropic microstructures. Since the voids could not be considered in our FFT homogenization due to infinite contrast between void properties, i.e., zero TCs, and other phases’ properties, we considered a two-step homogenization. In the first step, given the VFs of tungsten and voids, the same upper-bound properties were assigned to the tungsten voxels and the void voxels by considering all of them as a binary composite of pure tungsten and voids. In the second step, the numerical FFT homogenization method was applied to the whole RVE with the upper-bound property for the tungsten and void phases. Due to the statistical variations in our computational methods of microstructure generation and FFT homogenization, ten microstructures were generated and homogenized for each data point or void VF in the figure. Therefore, each data point represents the average of ten homogenized values.

## 4. Conclusions

This research introduced a new supervised ML strategy for expediting the prediction of bicontinuous heterogeneous composites’ TC and developing such composites with target properties. Our framework has a much faster computing speed than traditional numerical and optimization techniques. A big dataset encompassing the whole design space has been created for the tungsten-30% copper composite. Therefore, the focus of this study was to investigate the effect of microstructure morphologies on the properties. The microstructure realizations based on the DoE from this study can be applied to similar heterogeneous bicontinuous materials with varying constituents. In addition, a surrogate ML model was trained on the dataset to establish direct links between the microstructure and conductivity property and depict them using multiple response surfaces in minutes versus days via old methods of microstructure reconstruction and direct numerical solutions to homogenization problems. The VF is significantly more influential on the conductivity of the studied material system. As a result, we looked at the impact of defects on the composite’s microstructure and thermal properties. Finally, we were able to apply a modified autoencoder to explore the design space and locate the microstructural parameters of tungsten-30% copper composites resulting in desired properties in a few minutes.

## Figures and Tables

**Figure 1 materials-16-01088-f001:**
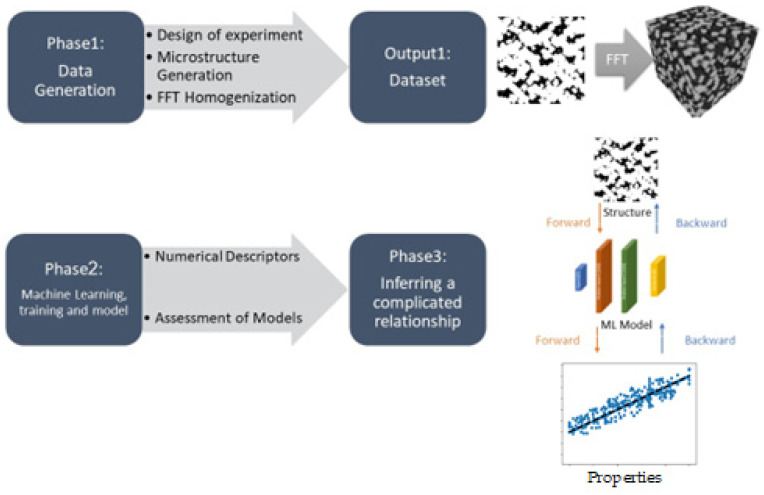
Overview of our study using an ML model for predicting the properties (forward links) and inverse design (backward links) of the studied material system, tungsten–copper composites.

**Figure 2 materials-16-01088-f002:**
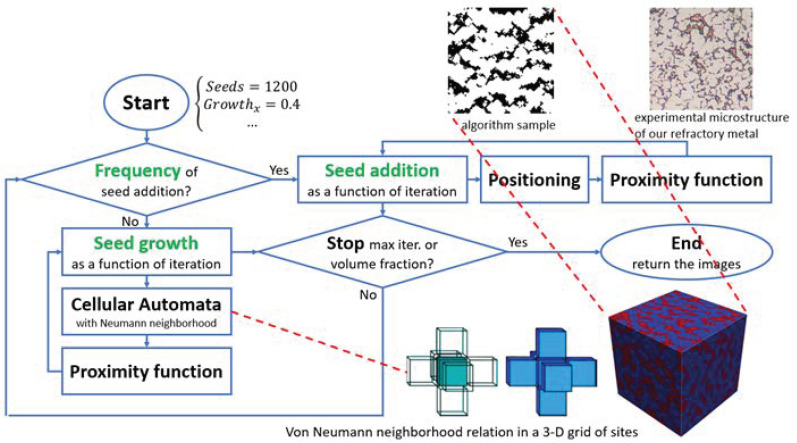
The flowchart of our algorithm generating artificial microstructures in silico.

**Figure 3 materials-16-01088-f003:**
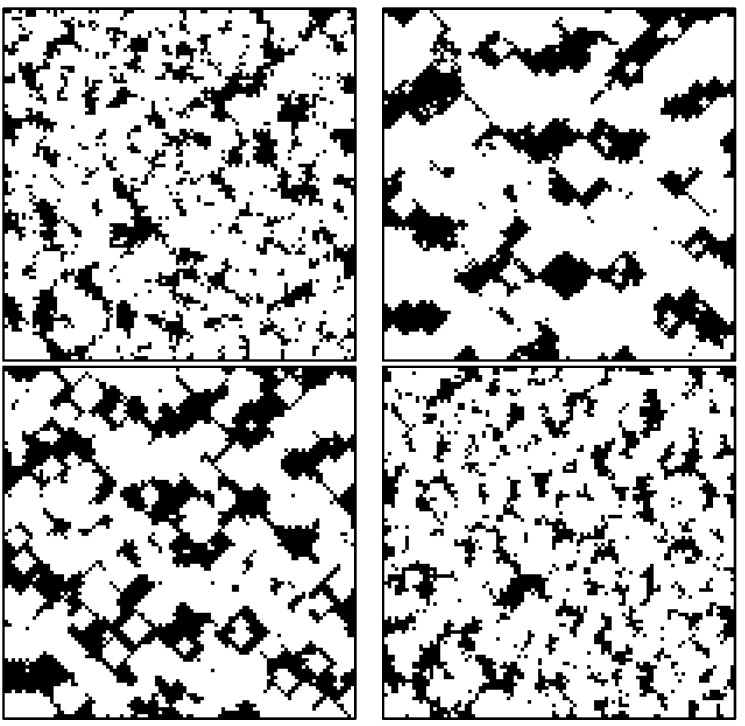
Several binary microstructures were made by changing the input parameters to test the efficiency of our realization code in generating morphologically diverse microstructures. Each 2D cut-section image above belongs to a distinct 3D microstructure, all having the same VF.

**Figure 4 materials-16-01088-f004:**
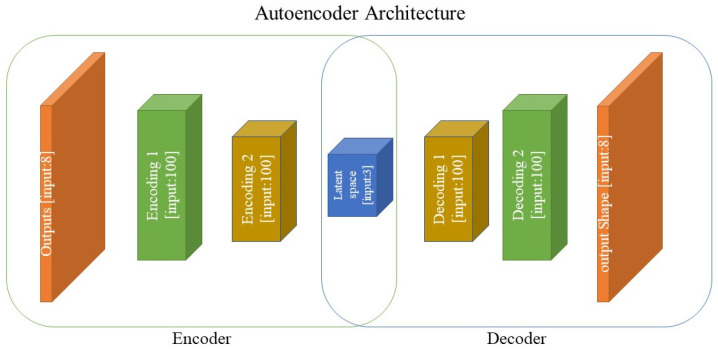
Modified autoencoder network architecture for fast material design and characterization.

**Figure 5 materials-16-01088-f005:**
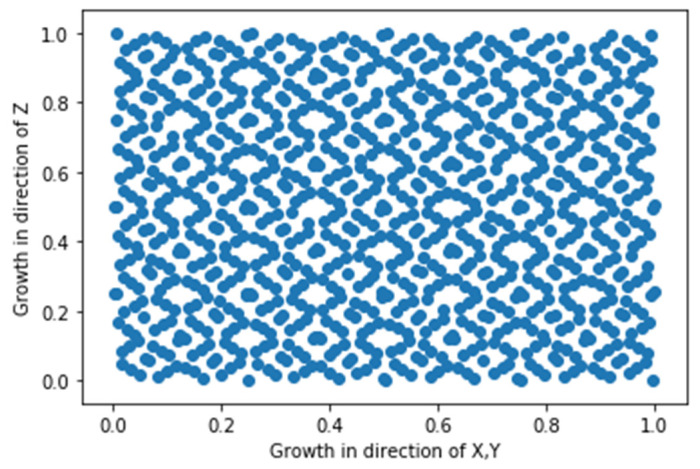
The 1000 feasible DoE points generated by the Sobol sequence shown on a 2D subspace of the 3D design space.

**Figure 6 materials-16-01088-f006:**
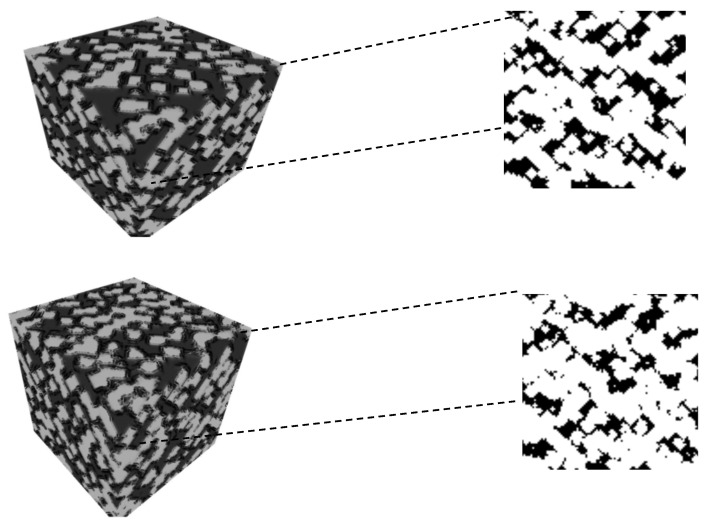
3D visualization for tungsten-30% copper with 100 voxels and different input variables.

**Figure 7 materials-16-01088-f007:**
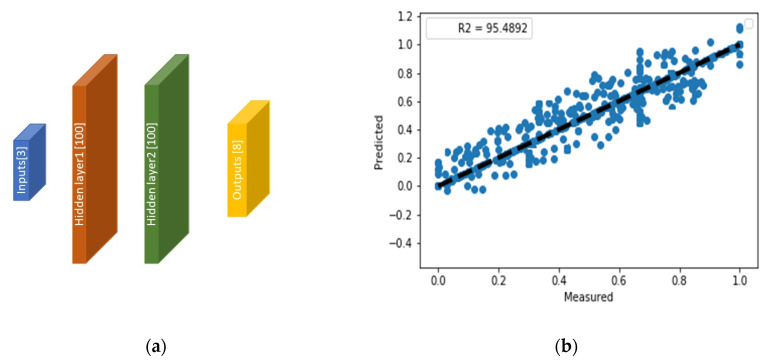
(**a**) The optimized fully connected neural network for direct structure–property links (surrogate of numerical homogenization) and (**b**) the network prediction’s regression plot (ordinate) with the ideal line of y=x.

**Figure 8 materials-16-01088-f008:**
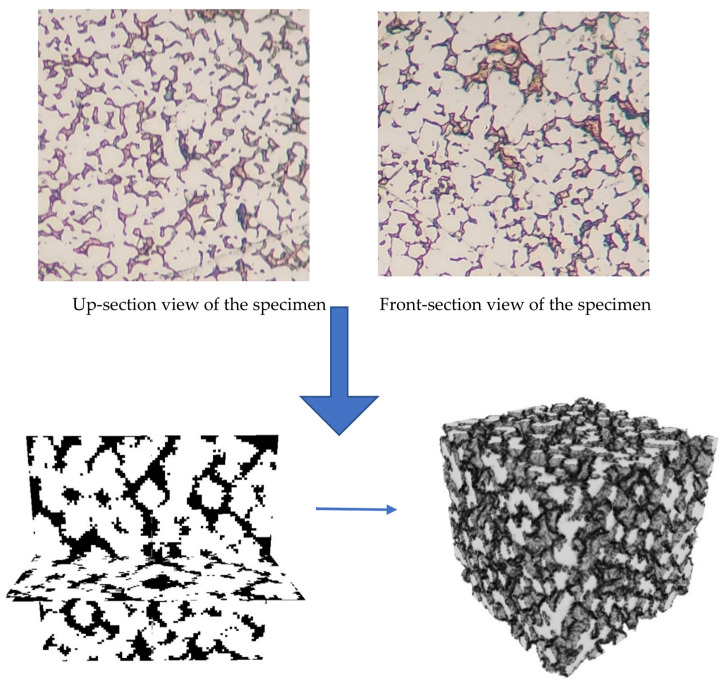
(**Top**) Perpendicular cut-section images of the experimental microstructure; (**Bottom**) the best match in our computational dataset found by the Figure 8 algorithm.

**Figure 9 materials-16-01088-f009:**
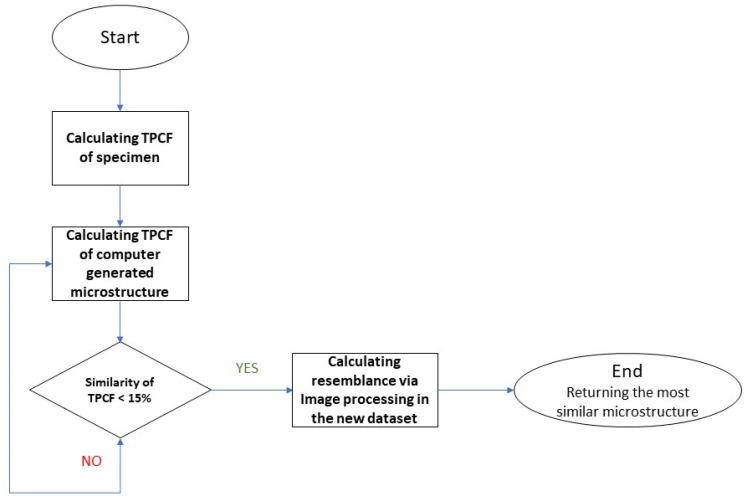
Flowchart of our algorithm for characterizing real or experimental microstructures using their two perpendicular images.

**Figure 10 materials-16-01088-f010:**
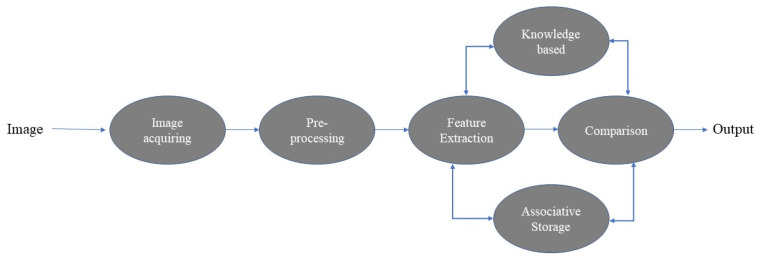
The high-level overview of our image processing algorithm.

**Figure 11 materials-16-01088-f011:**
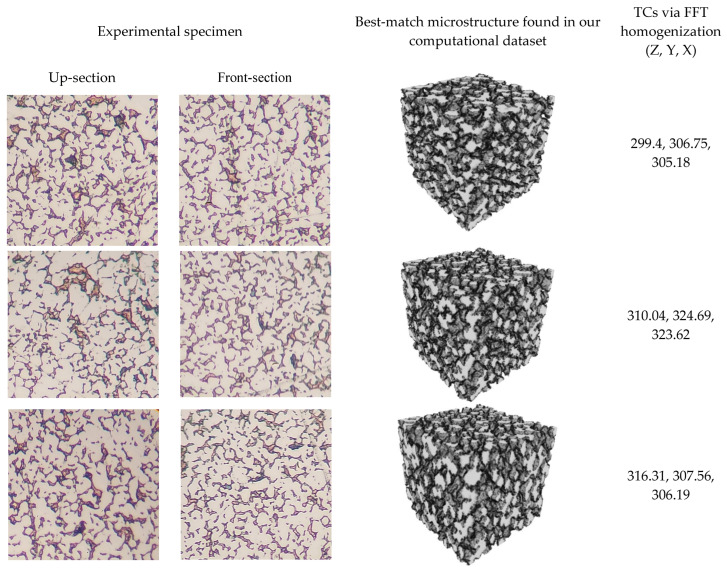
Estimated TCs of three experimental specimens using our method, searching for the computational microstructure similar to the given cut-section images of the specimen.

**Figure 12 materials-16-01088-f012:**
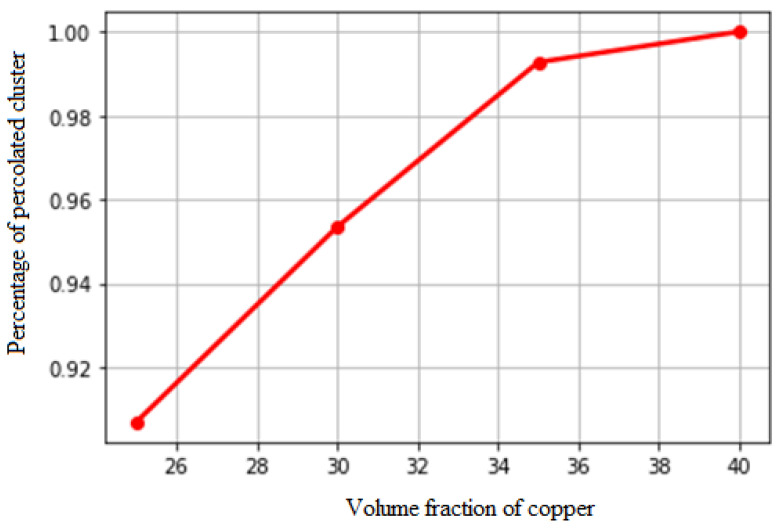
The effect of the copper VF on the largest cluster size is divided by the microstructure size (vertical axis).

**Figure 13 materials-16-01088-f013:**
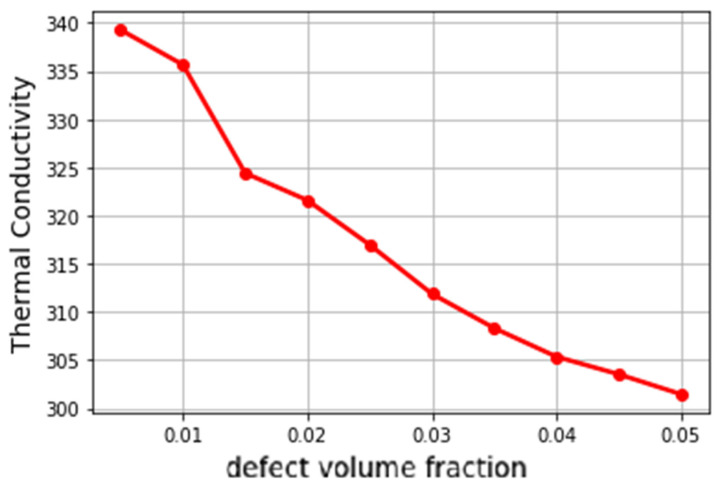
The effect of void percentage on the TC property in isotropic microstructures generated computationally.

**Table 1 materials-16-01088-t001:** Input variables of microstructure realization: the initial values for the test phase and the final values for generating the data set.

Input Parameters	Preliminary Test to Find the Optimal Values of Input Parameters	Exact Values and Range of Input Parameters after the Initial Test
Frequency of additional seeds	−1000–(−2000)	−1600–(−1900)
Growth rate	0.0001–1	0.0001–1
Initial seeds	50–10,000	2000
VF of tungsten	0.7	0.7

**Table 2 materials-16-01088-t002:** Average TC (among three diagonal components of TC tensor) compared between 100-voxel-per-dimension RVEs and their 200-voxel-per-dimension counterparts generated as described in Section 2.1.1.

Growth Probabilities in X and Y Direction (Equal) and Z Direction		Average TC (W/mK) with *N* = 200	Average TC (W/mK) with *N* = 100
0.5	0.5	327.18	334.41
0.75	0.25	308.15	307.31
0.25	0.75	322.45	321.52
0.375	0.375	317.83	316.83
0.875	0.875	311.89	311.76
0.625	0.125	327.05	326.97
0.125	0.625	311.81	310.61
0.1875	0.3125	317.23	317.19
0.6875	0.8125	302.66	300.98
0.9375	0.0625	313.66	312.26

**Table 3 materials-16-01088-t003:** Grid-searching neural network designs for ML hyperparameter optimization.

	Number of Neurons	Mean Square Error (MSE)
3-hidden layer	100, 50, 25	0.0295574
	100, 100, 50	0.0920061
	100, 100, 25	0.0531873
	100, 50, 50	0.0277496
	100, 25, 25	0.0671589
	100, 100, 100	0.0265248
	50, 50, 25	0.0278472
	50, 25, 25	0.0285401
	50, 50, 50	0.0675849
	25, 25, 25	0.02805
2-hidden layer	100, 100	0.0261359
	100, 50	0.0307351
	100, 25	0.0567823
	50, 50	0.0559963
	50, 25	0.0290273
	25, 25	0.0367758
1-hidden layer	100	0.0757515
	50	0.0397491
	25	0.0428257

**Table 4 materials-16-01088-t004:** Total or average computational times for various processes in minutes.

Realization of 1000 RVEs	FFT Homogenization/RVE	Surrogate Model Training and Prediction/RVE	Inverse Design
90	270	2	5

## Data Availability

Data is available upon request.

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
