# Peer review of "Design of Refractory Alloys for Desired Thermal Conductivity via AI-Assisted In-Silico Microstructure Realization"

_materials, 2023, doi:10.3390/ma16031088_

Round 1
Reviewer 1 Report
The manuscript “Design of refractory alloys for desired thermal conductivity via AI-assisted in-silico microstructure realization” contains the study of a computational methodology based on supervised machine learning. The tungsten-copper composite models were established and the thermal conductivity was analyzed by the trained ML model.
1. There are some spelling and grammatical errors in the manuscript. Such as, “desining” in line 52, “tungstan” in line 245, “Chacterization” in line 276, et al.
2. The second paragraph of the Introduction is too long to read. it is hard to find the main information of this paragraph and recommended to divide it into several paragraphs according to the content.
3. Check the reference citations, for example, there are no references in lines 79-83.
4. There are many names like tungsten-copper composite, tungsten copper composite (line 67), copper tungsten composite (line 85) and W-Cu composite (line 93) in the manuscript. It should be checked and unified in full manuscript.
5. Are the specimens in Fig.11 the same? If not, what is the difference between them? There should be more description.
6. Section “3.4.2. Volume fraction and defect effects” is mainly a description of the results, lacking the analysis and discussion of the reasons and mechanisms.
Author Response
Reviewer 1
Due to the decision of the journal about the necessity of a major revision and some language and clarity issues raised by the reviewers, we rewrote the whole manuscript firstly to make it more coherent with no language issues and secondly to include more explanations and corrected parts in response to valuable reviewers’ comments. We thank the editor and the reviewers for their constructive feedback.
Since we have addressed the issues in our revised manuscript, and many issues were due to the language used in our submitted manuscript, our responses are brief in this letter. The reviewer’s comments are in blue color in this letter.
The manuscript “Design of refractory alloys for desired thermal conductivity via AI-assisted in-silico microstructure realization” contains the study of a computational methodology based on supervised machine learning. The tungsten-copper composite models were established and the thermal conductivity was analyzed by the trained ML model.
- There are some spelling and grammatical errors in the manuscript. Such as, “desining” in line 52, “tungstan” in line 245, “Chacterization” in line 276, et al.
Thank you for your comment. We have fixed the spelling and grammatical errors, including the abovementioned ones, in our revised manuscript. You can refer to our revised manuscript for more information.
- The second paragraph of the Introduction is too long to read. it is hard to find the main information of this paragraph and recommended to divide it into several paragraphs according to the content.
We have divided the second paragraph in our revised manuscript per reviewer’s comment. We have also edited the text.
- Check the reference citations, for example, there are no references in lines 79-83.
We have added the appropriate reference for this part in our revised manuscript (this part has the same reference as the next sentence, i.e., [28]).
- There are many names like tungsten-copper composite, tungsten copper composite (line 67), copper tungsten composite (line 85) and W-Cu composite (line 93) in the manuscript. It should be checked and unified in full manuscript.
We have used consistent terminology in our revised manuscript.
- Are the specimens in Fig.11 the same? If not, what is the difference between them? There should be more description.
They are not the same. There were three experimental specimens which were mapped to their properties by our aforementioned TPCF-Image-Processing algorithm. The images are the 3D visualizations of the best-match microstructures found in our computational dataset. We have added better explanation about this figure in its above paragraph. Please refer to our revised manuscript for more details.
- Section “3.4.2. Volume fraction and defect effects” is mainly a description of the results, lacking the analysis and discussion of the reasons and mechanisms.
We have revised this section and added a better discussion about this part. Please refer to our revised manuscript for more details.
Reviewer 2 Report
This paper describes design of refractory alloys for desired thermal conductivity via AI-assisted in-silico microstructure realization. This paper fits the scope of the journal. However, some revisions are necessary before publication.
[1] Introduction
The second paragraph is too long, so please separate it.
[2] Line 52
The authors must revise the following sentence (Tungsten-?).
“In this study, we focus on desining particulate composites of Tungsten-.”
[3] Line 86
“We attempted to get the best qualities in manufacturing tungsten composites with 20% copper by weight in this study.”
Please cite the paper presenting the above result.
[4] Line 88
“In recent decades, there has been a lot of research into determining the functional qualities of composite materials based on their unique ingredients.”
In order to clearly show the novelty of this paper, the research history especially on the computational prediction should be described in detail citing papers. What is different from previous researches?
[5] Line 146
The abbreviation of “DoE” must be explained. Design of Experiment?
[6] Fig. 13
Fig. 13 is same as Fig. 12. The authors must check the figure and revise it as necessary.
Author Response
Reviewer 2
Due to the decision of the journal about the necessity of a major revision and some language and clarity issues raised by the reviewers, we rewrote the whole manuscript firstly to make it more coherent with no language issues and secondly to include more explanations and corrected parts in response to valuable reviewers’ comments. We thank the editor and the reviewers for their constructive feedback.
Since we have addressed the issues in our revised manuscript, and many issues were due to the language used in our submitted manuscript, our responses are brief in this letter. The reviewer’s comments are in blue color in this letter.
This paper describes design of refractory alloys for desired thermal conductivity via AI-assisted in-silico microstructure realization. This paper fits the scope of the journal. However, some revisions are necessary before publication.
[1] Introduction
The second paragraph is too long, so please separate it.
Thank you for your comment. We have fixed the language issues, including the above comment, in our revised manuscript. You can refer to our revised manuscript for more information.
[2] Line 52
The authors must revise the following sentence (Tungsten-?).
“In this study, we focus on designing particulate composites of Tungsten-.”
We have fixed the spelling and grammatical errors, including the above comment, in our revised manuscript.
[3] Line 86
“We attempted to get the best qualities in manufacturing tungsten composites with 20% copper by weight in this study.”
Please cite the paper presenting the above result.
We were referring to our study in the above sentence. We rewrote it in our revised manuscript for better clarity.
[4] Line 88
“In recent decades, there has been a lot of research into determining the functional qualities of composite materials based on their unique ingredients.”
In order to clearly show the novelty of this paper, the research history especially on the computational prediction should be described in detail citing papers. What is different from previous researches?
Introduction section has been edited to address this. As indicated there, the focus of this research was to investigate the effect of the microstructure of the composites (here refractory alloy) on its properties using ML models. Due to some unique features of this class of materials such as two percolated material phases, this work has yet to be done with this extension.
[5] Line 146
The abbreviation of “DoE” must be explained. Design of Experiment?
Yes, it is. The language issues, including this mistake, have been corrected in our revised manuscript.
[6] Fig. 13
Fig. 13 is same as Fig. 12. The authors must check the figure and revise it as necessary.
Figure 13 was incorrectly copied in our submitted manuscript. We have replaced it with the correct figure with relevant information. Please refer to our revised manuscript for more details.

Round 2
Reviewer 1 Report
After the revisions and the explanations made at the request of the two referees, the manuscript is now of good quality and can bring interesting information to the community of materials. Please check the language quality of the manuscript carefully.
Author Response
Thank you very much for your review. We have improved the language of the manuscripts.
Reviewer 2 Report
I have checked the revised version, but think that points 3 and 4 are not revised adequately.
Author Response
[3] Line 86
“We attempted to get the best qualities in manufacturing tungsten composites with 20% copper by weight in this study.”
Please cite the paper presenting the above result.
Answer: we have added two more references for this
Wang, Y., Zhuo, L. and Yin, E., 2021. Progress, challenges and potentials/trends of tungsten-copper (WCu) composites/pseudo-alloys: Fabrication, regulation and application. International Journal of Refractory Metals and Hard Materials, 100, p.105648.
Faraji, G., 2022. Fabrication of W-Cu composite by wire crumpling and subsequent melt infiltration. Materials Letters, 321, p.132432.
[4]“In recent decades, there has been a lot of research into determining the functional qualities of composite materials based on their unique ingredients.”
In order to clearly show the novelty of this paper, the research history especially on the computational prediction should be described in detail citing papers. What is different from previous researches?
the highlighted section has been added to the introduction to answer this question. relevant citations also have been added.